# A Scoping Review of Cognitive Bias in Internet Addiction and Internet Gaming Disorders

**DOI:** 10.3390/ijerph17010373

**Published:** 2020-01-06

**Authors:** Doris X.Y. Chia, Melvyn W.B. Zhang

**Affiliations:** 1Institute of Mental Health, National Addictions Management Service, Singapore 539747, Singapore; doris_xy_chia@imh.com.sg; 2Family Medicine & Primary Care, Lee Kong Chian School of Medicine, Nanyang Technological University, Singapore 308322, Singapore

**Keywords:** attention bias, cognitive bias, internet addiction, internet gaming disorder, psychiatry

## Abstract

Internet addiction and Internet gaming disorders are increasingly prevalent. Whilst there has been much focus on the use of conventional psychological approaches in the treatment of individuals with these addictive disorders, there has also been ongoing research exploring the potential of cognitive bias modification amongst individuals with Internet and gaming addiction. Some studies have documented the presence of cognitive biases and the effectiveness of bias modification for Internet addiction and gaming disorders. However, there have not been any reviews that have synthesized the findings related to cognitive biases for Internet addiction and Internet gaming disorders. It is important for us to undertake a scoping review as an attempt to map out the literature for cognitive biases in Internet addiction and gaming disorders. A scoping review was undertaken, and articles were identified using a search through the following databases: PubMed, MEDLINE, and PsycINFO. Six articles were identified. There were differences in the methods of ascertaining whether an individual has an underlying Internet or gaming addiction, as several different instruments have been used. With regards to the characteristics of the cognitive bias assessment task utilized, the most common task used was that of the Stroop task. Of the six identified studies, five have provided evidence documenting the presence of cognitive biases in these disorders. Only one study has examined cognitive bias modification and provided support for its effectiveness. Whilst several studies have provided preliminary findings documenting the presence of cognitive biases in these disorders, there remains a need for further research evaluating the effectiveness of bias modification, as well as the standardization of the diagnostic tools and the task paradigms used in the assessment.

## 1. Introduction

Internet addiction and Internet gaming disorders are increasingly prevalent. According to the International Classification of Diseases, ICD-11, gaming disorder refers to a maladaptive pattern of online or offline gaming [1]. It is characterized by the following symptoms: a loss of control over gaming; gaming taking precedence over other life priorities; and persistent or intensified gaming, despite there being negative consequences [1]. Whilst ICD-11 only recently introduced a diagnosis for Internet gaming, Internet addiction and Internet-associated addictive behaviors have been widely studied. Weinstein et al. [2] in their prior literature review reported the prevalence rates of Internet addiction to be between 1.5% and 8.2% in the United States and Europe. Müller et al. [3] separately examined the prevalence rate for Internet Gaming Disorder, and reported it to be between 1.6% and 5.1%, based on a sampled cohort of 12,938 adolescents. These figures highlight that these disorders are highly prevalent. To date, there have not been any approved pharmacological medications available to help individuals abstain from excessive Internet use or Internet gaming. The mainstay of treatment remains that of psychological interventions, given that these disorders are behavioral conditions in nature. It is of importance for there to be effective interventions targeting all the factors leading individuals to develop addictive disorders, given that these disorders are most commonly found amongst adolescences, and the adolescent brain is still developing. In addition, previous research has notably reported an association between personality and smoking [4]. To date, several different psychological treatment modalities have been considered, most commonly that of cognitive behavioral therapy, and other modalities such as family therapy and reality therapy [5]. Zajack et al. [5] have attempted, in their prior review, to determine which of the above-mentioned psychological interventions was most effective. However, they did not reach a conclusion. Whilst there has been much focus on the use of conventional psychological approaches in the treatment of individuals with these addictive disorders, there has also been ongoing research exploring the potential of cognitive bias modification amongst individuals with Internet and gaming addiction.

The advances in experimental psychology in the last decade has led to a more in-depth understanding of cognitive biases. Cognitive biases include that of attentional, approach, and interpretational tendencies, and these automatic processes could cause individuals to attend more readily to stimulus in their environment, such as substance-related cues or threatening stimuli [6]. By attending more readily to these cues, this results in a relapse back into their addictive disorder or results in heightened anxiety. These automatic processes have been implicated for several psychiatric disorders, ranging from that of addictive disorders, such as alcohol use and opioid use, to that of anxiety and even depressive disorder [6,7,8]. It is postulated that conventional psychological approaches have addressed the conscious cognitive control processes, but not underlying unconscious processes [9]. These unconscious processes would result in individuals with addictive disorders orientating preferentially towards substance-related cues in the environment, and those with anxiety disorders to attend more readily to threatening stimuli [9]. Given the presence of these biases, bias modification interventions are now available and have been widely evaluated. Cristea et al. [10] in their prior review highlighted that cognitive bias modification was moderately effective in addressing automatic biases amongst individuals with an addictive disorder, with an effect size of 0.60. Jones et al. [6] have also in their review of meta-analyses reported that bias modification helped in ameliorating attention bias and interpretative biases, with effect sizes of 0.24–1.16 and 0.52–0.81, respectively. There has also been evidence to suggest that these cognitive biases are present in behavioral addictions, such as that of gambling disorders. Hønsi et al. [11], in their systematic review that synthesized the findings from eleven studies, reported that both individuals with problem and pathological gambling had underlying attentional biases towards gambling-related cues. Attentional biases were also consistently detected, despite there being a variety of different measurement tools utilized. More recently, McGrath et al. [12] utilized eye-gaze tracking and demonstrated that individuals with gambling disorders tend to attend to cues that are specific to their original type of gambling. Further work has been reported by Boffo et al. [13] in their protocol, which documented their intent to investigate the effectiveness of modifying both attentional and approach biases for individuals with gambling problems.

For Internet addiction and Internet gaming disorder, there has been preliminary work undertaken in the exploration of the presence of cognitive biases amongst individuals with these disorders. Jeromin et al. [14] utilized an addiction Stroop task in order to examine whether excessive Internet gamers displayed attentional biases to general computer cues. Dong et al. [15] have also previously investigated whether individuals with addictive disorders had impaired executive control by using the classic color-word Stroop task. From our knowledge to date, there have not been any reviews that have synthesized the findings relating to cognitive biases for Internet addiction and Internet gaming disorders. It is important for us to undertake a scoping review as an attempt to map out the literature for cognitive biases in Internet addiction and gaming disorders. Scoping reviews have been increasingly popular as an approach for synthesizing research evidence for areas of research in which the topic of interest has not yet been extensively researched into, or is heterogeneous in nature [16]. By undertaking a scoping review, we seek to better understand the extent and range of research that has been undertaken for cognitive biases in Internet addiction and gaming disorders, to help in the identification of gaps in the research literature, and to inform the steps required for a systematic review.

## 2. Methods

### 2.1. Data Sources and Search Strategy

For the purposes of this scoping review, articles were identified using a search through the following databases: PubMed, MEDLINE, and PsycINFO. The following search terminologies were used: (“attention bias” OR “cognitive bias” OR “approach bias” OR “avoidance biases”) AND (“Internet addiction” OR “Internet Gaming Disorder” OR Computer addiction” OR “Video Gaming”). The search strategy was modified to suit the different databases, in consultation with a librarian at the Lee Kong Chian School of Medicine, Nanyang Technological University Singapore. Table 1 provides an overview of the search strategies used for each of the databases. All the databases were searched from inception through to 28 March 2019.

### 2.2. Inclusion and Exclusion Criteria

Articles that were written in the English language were included. Articles were included if (1) the condition examined was that of Internet addiction or Internet gaming Disorder or Video Gaming, (2) it was stated explicitly that cognitive biases were assessed for or modified. Articles were excluded if (1) the condition examined was any other psychiatric conditions, (2) there was no mention of any assessment or modification of cognitive biases.

### 2.3. Data Extraction

The following information was extracted from each of the articles: (1) publication details (names of the authors and the year of publication), (2) the sample size in the studies, (3) the study design, (4) the diagnosis of the participants, (5) the cognitive bias task used, and (6) the outcomes of cognitive bias assessment or modification. Data was extracted from each of the identified articles and recorded on an electronic spreadsheet by one of the authors (D.C.) and cross-checked by another author (M.Z.). Figure 1 provides an overview of the selection process of the articles, in which data was extracted systematically.

### 2.4. Analysis

A narrative qualitative synthesis was performed, given that this was a scoping review, and the main objective was to map out the research done in the field.

## 3. Results

### 3.1. Findings

A total of 70 articles were identified in accordance with the search strategy. A total of 16 articles were duplicates. The titles and the abstracts of these papers were inspected, and only 15 full-text articles were then assessed against the inclusion and exclusion criteria. Six articles were included for the qualitative synthesis.

### 3.2. Characteristics of Included Studies

Table 2 provides an overview of the characteristics of the included studies (*n* = 6). All the studies were cross-sectional in study design. In terms of the aims of the studies, all the studies attempted to determine the existence of attentional biases, except for one study which also attempted to determine the effectiveness of cognitive bias modification [17]. Two out of the six studies have involved participants from an Asian cohort, that of China [15,18]. The methods of screening for the presence of Internet addiction or Internet gaming disorder varied, with different questionnaires used. The questionnaire used included that of the Young’s Internet Addiction Test, Game Addiction Scale, the Modified Diagnostic Questionnaire for Internet Addiction, Compulsive Internet Use Scale for World of Warcraft (WoW), and that of the Internet Gaming Disorder Questionnaire.

### 3.3. Characteristics of the Cognitive Bias Assessment Tools Utilized

The Stroop task was the most commonly used method for the assessment of bias, with four out of the six assessed studies utilizing it. In the Stroop task, participants are asked to indicate the font color of each word by pressing on the appropriate keys. Other methods of assessment include the use of the dot-probe task, the Internet game-shifting task, and the modified gaming version of the approach-avoidance task. For the dot-probe task, participants are asked to identify the position of a probe, that appears in the position of either the stimulus or neutral image. For the Internet game-shifting task, either computer-related or neutral images are specified as targets and participants are required to indicate a response as quickly as possible when a target is shown. For the approach-avoidance task, participants are asked to pull or push triggering stimuli and neutral stimuli.

There remains to be variation in the nature of the stimulus used, even though the same task (for example that of the Stroop task) was used. For example, in Dong et al. [15]’s study, words of different colors were used. Meanwhile in van Holst et al. [19]’s study, game-related and movie-related words, which were matched in terms of length and phonetics, were used. In two other studies, computer-related and office-related words were used instead [14,20]. There were variations in the task paradigm across the studies.

### 3.4. Evidence for the Cognitive Bias and Bias Modification

All of the studies, except for Jeromin et al. [20] provided evidence for the presence of attentional biases. The only study that have examined the evidence for cognitive bias modification also reported it to be effective in reducing attentional biases [17].

## 4. Discussion

This study is perhaps the first study that has reviewed the existing literature of cognitive biases amongst individuals with Internet addiction and Internet gaming disorder. This review is timely, given that the ICD-11 has just recently proposed for the official inclusion of gaming disorder into their diagnostic and classification system. The aim of this review was largely to scope and map out the work done in the field. Based on the identified studies, it is telling that there has been an extensive evaluation of cognitive biases in these disorders, with the first of these studies being published as early as 2011. This review also highlighted the diversity in the methods of ascertaining if an individual has an underlying Internet or gaming addiction, as several different instruments have been used. With regards to the characteristics of the cognitive bias assessment task utilized, the most common task used was the Stroop task. Of the six identified studies, five have provided evidence documenting the presence of cognitive biases in these disorders. One study has reported that bias modification was effective in reducing bias.

Most of the studies (five out of the six) identified have reported the presence of cognitive biases in their sampled participants. This is notable despite there being differences in the diagnostic questionnaire used and the variability in the methods of assessment of cognitive biases. This finding is congruent with the previous findings by Hønsi et al. [11], which after synthesizing the findings from eleven separate studies in their review, reported that individuals with gambling addiction have attentional biases towards gambling-related cues. Internet addiction and Internet gaming disorders are forms of behavioral addiction, and the presence of these biases highlight that there remains a need for interventions to modify and ameliorate these biases. In our literature review, we have highlighted Zajac et al. [5]’s prior review, which concluded that there is inconclusive evidence to recommend any particular types of psychological therapy. However, in Zajac et al. [5]’s prior review, the methods of psychological interventions were mainly limited to that of cognitive behavioral approaches. Therapies like cognitive behavioral therapies might be effective in modifying underlying cognitive distortions, and maladaptive thinking, as well as the conscious cognitive control process. However, there is a need for interventions that could help with these automatic, unconscious processes, such as attentional and approach biases, given that existing therapies do not typically deal with these automatic processes. Moreover, Rabinovitz et al. [17]’s prior study has demonstrated that these automatic processes could be modified. Modification of these biases could potentially help in improving these underlying addictive behaviors. Cognitive bias modification has been extensively studied for addictive disorders, and Cristea et al. [10], in their prior review, have reported that there was a moderate effect size of cognitive bias modification for substance (alcohol and tobacco) addictions. To date, there have also been a series of studies conducted in the clinical settings for alcohol use disorders [21,22,23] that have provided robust evidence on the effectiveness of bias modification. Thus, there remains a need for further research examining the potential of modifying cognitive biases amongst individuals with Internet addiction and gaming disorders.

The Stroop task was the most commonly used task for the assessment of cognitive biases. However, several other tasks were also used, including that of the dot-probe task, the Internet game-shifting task, and the modified gaming version of the approach-avoidance task. In this review, we have attempted to extract as much information as we could, based on what has been described in the methods of each of the articles, with regards to the nature of the task that was utilized. From the evidence synthesized, there remains major variability in the nature of the tasks, even if the same task was used. This is congruent with what Zhang et al. [24] have reported in their evaluation of the visual probe task paradigm for individuals with substance use disorders. In Zhang et al. [24]’s prior review, they reported variations in the timing, the number of included trials, the timings for the fixation cross, and the timings of the stimulus set. In this review, we found great variability not only in the nature of the task, but also in the nature of the stimulus that was used. The variations in the task paradigms are not helpful for future research, as the differences in the task paradigm affect the reproducibility of the results, and the reliability of the task. A variety of stimulus words or images were used, with some related to the genre of games that the individual was playing, whilst others were non-specific and just related to computer items. Prior research has recommended for the personalization of the stimulus, and this should be considered in future research. Participants should be recruited and asked to rate images or words for relevance, before they are to be used in the intervention.

There are several research implications that arise from this existing review. The fact that cognitive biases are present in some of the identified studies suggests a need for a more extensive evaluation of the presence of these biases amongst individuals with Internet disorders, and an evaluation of the effectiveness of bias modification. Moreover, future research should also seek to standardize the diagnostic methods for ascertaining whether an individual has an Internet-related disorder. Most of the current research have relied on questionnaire-based assessment of whether an individual has obtained an underlying diagnosis of Internet addiction or gaming disorder. It might be ideal for future research to consider using either a clinician-rated questionnaire or having a clinician make a diagnosis based on a psychiatric interview. It is also of importance for future research to standardize the paradigm of the cognitive bias modification intervention and to report the nature of the task paradigm fully, so as to allow for reproducibility in other settings and in other studies. Researchers should also consider asking participants to personalize the stimulus images or words used, so that they are more of relevance to them, as this would affect the magnitude of the biases detected and the effectiveness of modification.

The strengths of this review are that this is perhaps the first review that has systematically mapped out the evidence for cognitive biases in Internet addiction and Internet gaming disorder. A comprehensive search was conducted on several databases. Articles were also screened against a set of inclusion and exclusion criteria. However, several limitations remain. Whilst our search was comprehensive, we acknowledge that we might have missed some articles, as this is a rapidly emerging field of research, given the recent official inclusion of gaming disorder into the ICD-11. We have also excluded articles that examined cognitive biases by using functional imaging. Given that this is a scoping review, we are limited to a qualitative discussion of the results from the articles identified.

## 5. Conclusions

This article has reviewed and scoped out the extent of research into cognitive biases for Internet addiction and Internet Gaming Disorders. Several studies have provided evidence for the presence of cognitive biases amongst individuals with Internet addiction and gaming disorders, while one study has demonstrated the effectiveness of bias modification. The results arising from this scoping review highlight a need for future research to more thoroughly evaluate the effectiveness of bias modification. It is also important for future research to consider the use of standardized questionnaires for the diagnosis of these disorders, and for researchers to adhere to a standard paradigm for cognitive bias assessment and modification. This is of importance for future synthesis of the overall effectiveness of bias modification, and for future clinical application.

## Figures and Tables

**Figure 1 ijerph-17-00373-f001:**
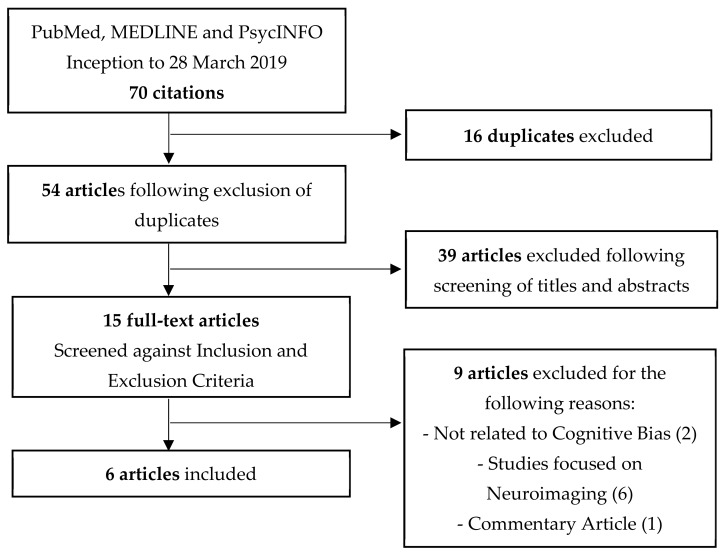
Overview of the Selection Process of the Articles.

**Table 1 ijerph-17-00373-t001:** Search Strategies used.

Database	Search Strategy
PubMed	Search (“Internet”[Mesh]) AND “Behavior, Addictive”[Mesh]
	Search (internet addiction[Title/Abstract]) OR internet addiction[Text Word]
	Search ((internet gaming disorder[Title/Abstract] OR computer addiction[Title/Abstract])) OR (internet gaming disorder[Text Word] OR video gaming[Text Word])
	Search ((attention bias[Title/Abstract]))
	Search ((cognitive bias[Title/Abstract]))
	Search ((approach bias[Title/Abstract]))
	Search ((avoidance bias[Title/Abstract]))
MEDLINE	(1) Attentional Bias OR (cognitive bias OR attention * bias OR avoidance bias OR approach bias) ab,ti,tw.
	(2) (Internet gaming disorder OR Internet addiction OR computer addiction OR video gaming) ab,ti,tw.
	(3) 1 and 2
PsycINFO	AB internet addiction OR TI internet addiction
	TI (internet gaming disorder OR computer addiction) OR AB (internet gaming disorder OR computer addiction)
	DE “Attention Bias”
	DE “Cognitive Bias”
	DE “Approach bias”
	DE “Avoidance Bias”

Note: *: Truncation; AB—Abstract; TI—Title; DE—Descriptors.

**Table 2 ijerph-17-00373-t002:** Overview of the Characteristics of the Included Studies.

Study	Study Design	Scope of Study	Participants	Method of Screening	Assessment Task	Nature of Stimulus Included	Details of Assessment Task	Findings
Dong et al. (2011) [15]	Cross-sectional	Presence of attentional bias	17 male participants with Internet Addiction Disorder (IAD) (mean age = 21.09 years old) 17 male control participants (mean age = 20.78 years old)	Young’s Internet Addiction Test (IAT)	Classic color-word Stroop Task	Color-related words (i.e., red, green, blue, and yellow)	Fixation Cross Timing: 250 ms Stimulus Timing: 600 ms Inter Stimulus Interval: 1000 ms Total number of critical trials: 240 Total number of practice trials: 40	Impaired executive control shown in individuals with IAD
van Holst et al. (2012) [19]	Cross-sectional	Presence of attentional bias	92 male adolescents (mean age = 15.1 years old)	Game Addiction Scale (GAS)	Dot-probe Task, Addiction-Stroop Task, and Go/no-go Task	Dot-probe Task: Screenshot pictures from popular video games vs. neutral cartoon pictures	Dot-probe Task: Fixation Cross Timing: Not mentioned Stimulus Pair Timing: 500 ms Probe Timing: 200 ms Inter-trial Interval: Not mentioned Stimulus Onset Asynchrony (SOA): Not mentioned Total number of actual trials: 100 Total number of practice trials: 10	Higher levels of video-gaming were associated with higher levels of attentional bias and response disinhibition
						Addiction-Stroop Task: Game-related words vs. movie-related words	Addiction-Stroop Task: Total number of critical trials: 51 Total number of neutral trials: 51	
						Go/no-go Task: (i) Basic Inhibition Condition: 120 animal pictures and 40 human pictures (ii) Game Condition: 120 car pictures and 40 game-related pictures	Go/no-go Task: Total number of critical trials: 40 for both conditions Total number of neutral trials: 120 for both conditions	
Zhou et al. (2012) [18]	Cross-sectional	Presence of cognitive bias	46 participants with Internet Game Addiction (IGA) (mean age = 26 years old; 69.6% males) 46 control participants (mean age = 26 years old; 69.6% males)	Modified Diagnostic Questionnaire for Internet Addiction (YDQ)	Internet Game-shifting Task	10 game-related pictures and 10 neutral fruit pictures.	Stimulus Timing: 500 ms Inter-trial Interval: 800 ms Total number of critical trials: 160 Total number of practice trials: 40	Individuals with IGA displayed cognitive bias and impaired executive functioning
Rabinovitz et al. (2015) [17]	Randomized Controlled Trial	Presence of approach bias and effectiveness of a single session Cognitive Bias Modification (CBM)	38 excessive multiplayer online male gamers (EG) randomly assigned to one training group: 19 avoidance training index group (mean age = 22.5 years old) 19 approach training control group (mean age = 23.1 years old)	Game Addiction Scale (GAS) and gaming hours per week	Modified gaming version of the Approach Avoidance Task (AAT)	Game-related pictures and cartoon pictures	AAT with four sequential phases Number of trials: Practice Phase—20 Pre-training Assessment—80 Training Phase—440 Post-training Assessment—80	EG showed approach bias to game cues. The single session CBM was effective in reducing approach bias such that the mean reaction time decreased significantly from pre to post approach for the avoidance training group but the opposite was found for the approach training group
Jeromin et al. (2016) [14]	Cross-sectional	Presence of attentional bias	21 excessive Internet gamers (mean age = 22.9 years old; 81% males) 30 non-gamers (mean age = 24.5 years old; 63.3% males)	German version of the Compulsive Internet Use Scale for WoW (CIUS-WoW)	Addiction Stroop and Visual Probe	Addiction Stroop: 20 computer-related words and 20 office-related words	Addiction Stroop: Stimulus Timing: Until a response key was pressed. Fixation Cross Timing: 1000 ms Total number of critical trials: 320 (5-min break between two blocks of 160 trials each). Total number of practice trials: 40	The findings from the Addiction Stroop suggested the presence of attentional bias in excessive gamers. No significant findings were found from the Visual Probe Test.
						Visual Probe: 10 computer-related pictures and 10 neutral pictures (e.g., radio)	Visual Probe: Fixation Cross Timing: Throughout the task. Stimulus Pair Timing: 150 or 450 ms Probe Timing: 200 ms Inter-trial Interval: 1000 or 2000 ms Stimulus Onset Asynchrony (SOA): 50 ms Total number of actual trials: 200 Total number of practice trials: 6	
Jeromin et al. (2016) [20]	Cross-sectional	Presence of attentional bias	Study 1: 27 gamers with Internet Gaming Disorder (IGD) (mean age = 24.9 years old; 70.4% males) 27 casual gamers (mean age = 28.3 years old; 70.4% males) 27 non-gamers (mean age = 31.2 years old; 70.4% males)	Study 1: German version of the Compulsive Internet Use Scale (CIUS)	Study 1: Web-based Addiction Stroop with randomised word design.	Study 1: 20 computer-related and 20 office-related words	Study 1: Duration of trial: Max. 1000 ms Stimulus Timing: Not mentioned. Fixation Cross Timing: Not mentioned. Total number of critical trials: 320 (self-timed break between two blocks of 160 trials each). Total number of practice trials: 40	The findings did not support the presence of attentional bias in individuals with IGD
			Study 2: 29 IGD male gamers (mean age = 23.3 years old) 29 casual male gamers (mean age = 23.3 years old) 29 male non-gamers (mean age = 23.5 years old)	Study 2: German version of the Internet Gaming Disorder Questionnaire (IGDQ)	Study 2: Web-based Addiction Stroop and Classical Stroop, both with block word design.	Study 2: Addiction Stroop—computer-related words and office-related words Classical Stroop—Color-related words (i.e., red, blue, green, and yellow) and numerical words (i.e., zero, five, nine, and eleven)	Study 2: Addiction Stroop with Block Design: Duration of block: Max. 48 s Stimulus Timing: Not mentioned. Fixation Cross Timing: Not mentioned. Total number of critical trials: 192 Total number of practice trials: Not mentioned. Classical Stroop with Block Design: Procedure: Same as that of Study 1 Duration of block: Max. 48 s Stimulus Timing: Not mentioned. Fixation Cross Timing: Not mentioned. Total number of critical trials: 192 Total number of practice trials: Not mentioned.	

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
