# Peer review of "A Scoping Review of Cognitive Bias in Internet Addiction and Internet Gaming Disorders"

_ijerph, 2020, doi:10.3390/ijerph17010373_

Round 1

Reviewer 1 Report

Thank you for sending me this manuscript for review. The theme is novel and interesting. I believe that the authors should make an effort to better defend the potential of the manuscript, especially emphasizing the usefulness and applicability of the results obtained, in relation to the possibilities of analysis offered and also in favor of the design of future lines of research, for intervention in Addiction & Internet Gaming Disorders.

Review and modify the wording of the ABSTRACT, it has a confusing reading.

In the INTRODUCTION I should further develop some issues related to risk behaviors, especially when referring to the adolescent stage, which is a period characterized by continuous changes and therefore a vulnerable group. For example, they could cite works such as: Pérez-Fuentes MC, Gázquez JJ, Molero MM, Cardila F, Martos Á, Barragán AB, Garzón A, Carrión JJ, Mercader I. Adolescent impulsiveness and use of alcohol and tobacco. European Journal of Investigation in Health, Psychology and Education. 2015; 5 (3): 371-382.

In the METHOD, they must specify whether a time limit for the search and selection of publications has been specified. In addition, it would be interesting to show specifically the search formulas used in each database consulted. In these cases, it may be appropriate to present this data in a table.

The publication selection process should be detailed in the Method section, not results.

The RESULTS table contains too much text. They must synthesize the information in the table, and complete the data by writing the text.

The CONCLUSIONS section is scarce. The authors mention some limitations superficially. However, there is a lack of argumentation about the usefulness and application of the results obtained, and also the proposal for future lines of research.

Author Response

We like to thank Reviewer 1 for recognizing the importance of our work. Please find as enclosed our in-line replies to your comments.

Review and modify the wording of the ABSTRACT, it has a confusing reading.

We have amended the abstract. This is the revised abstract:

Abstract: Internet addiction and Internet gaming disorders are increasingly prevalent. Whilst there has been much focus on the use of conventional psychological approaches in the treatment of individuals with these addictive disorders, there has also been ongoing research exploring the potential of cognitive bias modification amongst individuals with Internet and gaming addiction. Some studies have documented the presence of cognitive biases and the effectiveness of bias modification for Internet addiction and gaming disorders. However, there have not been any reviews that have synthesized the findings relating to cognitive biases for Internet addiction and Internet gaming disorders. It is important for us to undertake a scoping review as an attempt to map out the literature for cognitive biases in Internet addiction and gaming disorders. A scoping review was undertaken, and articles were identified using a search through the following databases (PubMed, MEDLINE and PsycINFO). Six articles were identified. There was differences in the methods of ascertaining if an individual has an underlying Internet or gaming addiction, as several different instruments have been used. With regards to the characteristics of the cognitive bias assessment task utilized, the most common task used was that of the Stroop task. Of the six identified studies, five have provided evidence documenting the presence of cognitive biases in these disorders. Only one study has reported that bias modification was effective in reducing bias. Whilst several studies have provided preliminary findings documenting the presence of cognitive biases in these disorders, there remains a need for there to be further research evaluating the effectiveness of bias modification, and the standardization of the diagnostic tools and the task paradigms used in the assessment.

In the INTRODUCTION I should further develop some issues related to risk behaviors, especially when referring to the adolescent stage, which is a period characterized by continuous changes and therefore a vulnerable group. For example, they could cite works such as: Pérez-Fuentes MC, Gázquez JJ, Molero MM, Cardila F, Martos Á, Barragán AB, Garzón A, Carrión JJ, Mercader I. Adolescent impulsiveness and use of alcohol and tobacco. European Journal of Investigation in Health, Psychology and Education. 2015; 5 (3): 371-382.

We thank you Reviewer 1 for highlighting this point. We have included this citation. The amends are as follows:

“It is of importance for there to be effective interventions targeting all the factors leading individuals to develop an addictive disorders, given that these disorder are most commonly amongst adolescences, and the adolescent brain is still developing; and also previous research has also reported that the association between personality and smoking [24].”

In the METHOD, they must specify whether a time limit for the search and selection of publications has been specified. In addition, it would be interesting to show specifically the search formulas used in each database consulted. In these cases, it may be appropriate to present this data in a table.

We seek to clarify that the databases were searched from inception up until the 28th of March 2019. We have already specified this in our manuscript. We have presented an overview of the search strategy for each of the databases in Table 1. Please note that the search was conducted based on the combination of individual searches.

Table 1. Search Strategies used

Database

Search Strategy

PubMed

Search ("Internet"[Mesh]) AND "Behavior, Addictive"[Mesh]

Search (internet addiction[Title/Abstract]) OR internet addiction[Text Word]

Search ((internet gaming disorder[Title/Abstract] OR computer addiction[Title/Abstract])) OR (internet gaming disorder[Text Word] OR video gaming[Text Word])

Search ((attention bias[Title/Abstract]))

Search ((cognitive bias[Title/Abstract]))

Search ((approach bias[Title/Abstract]))

Search ((avoidance bias[Title/Abstract]))

MEDLINE

1     Attentional Bias/ or (cognitive bias or attention* bias or avoidance bias or approach bias).ab,ti,tw.

2     (Internet gaming disorder or Internet addiction or computer addiction or video gaming) ab,ti,tw.

3     1 and 2

PsycINFO

AB internet N3 addiction OR TI internet N3 addiction

TI ( internet gaming disorder OR computer addiction ) OR AB ( internet gaming disorder OR computer addiction )

DE "Attention Bias"

DE "Cognitive Bias"

DE "Approach bias"

DE "Avoidance Bias"

The publication selection process should be detailed in the Method section, not results.

Thank you for highlighting this. We have moved this to be under 2.3

The RESULTS table contains too much text. They must synthesize the information in the table, and complete the data by writing the text.

We have modified the results table and summarized the main information in the table.

The CONCLUSIONS section is scarce. The authors mention some limitations superficially. However, there is a lack of argumentation about the usefulness and application of the results obtained, and also the proposal for future lines of research.

We have amended the conclusions to state the implications of the findings of our review and how it will affect future research.

The amends are as follows:

“This article has reviewed and scoped out the extent of research into cognitive biases for Internet addiction and Internet Gaming Disorders. Several studies have provided evidence for the presence of cognitive biases amongst individuals with Internet addiction and gaming disorders; and one study has demonstrated the effectiveness of bias modification. The results arising from this scoping review highlights there being a need for future research to evaluate more thoroughly the effectiveness of bias modification. It is also of importance for future research to consider the use of standardized questionnaires for the diagnosis of these disorders; and for them to adhere to a standard paradigm for cognitive bias assessment and modification. This is of importance for future synthesis of the overall effectiveness of bias modification, and for future clinical application.”

Reviewer 2 Report

It is my pleasure to read this interesting review. Which provides a scope to understand the cognitive bias of internet addiction. I had some suggestions to improve the paper. 

1 It is necessary to have a good discussion for the definition of cognitive bias. 

2 I agree that the way to find the article. However, as the terminology was varied for the cognitive bias. I will suggest extending the screening to find more articles. Actually, as I know, there are two articles had evaluated the memory bias and bias in response in response inhibition. (Kaohsiung J Med Sci. 2014 Jan;30(1):43-51. doi: 10.1016/j.kjms.2013.08.005. Epub 2013 Sep 14; Asia Pac Psychiatry. 2015 Jun;7(2):143-52. doi: 10.1111/appy.12134. Epub 2014 May 27.) It is not necessary to include these two articles. But, it might indicate some articles will miss in the current search way. 

3 It is good to have a table. However, the information in the table is too much. I will suggest summarising the information in the table.  It will make it more readable. 

4 It is good to have a suggestion for further study. However, the suggestion should be specified. For example, it is really problems that most studies confirm the diagnosis by questionnaire. Thus, it should be discussing what problems with the diagnosis. Then, it should suggest which way to make a diagnosis is adequate. For example, how to choice diagnosis by DSM-5 or ICD-11. 

Author Response

We thank you reviewer 2 for your positive comments and for your recommendations as to how our paper could be improved.

1 It is necessary to have a good discussion for the definition of cognitive bias. 

We have expanded the definition of cognitive bias in the introduction. The amends are as follows:

“Cognitive biases include that of attentional, approach and interpretational tendencies, and these automatic processes could cause individuals to attend more readily to stimulus in their environment, such as substance-related cues or threatening stimuli [5]. By attending more readily to these cues, this results in a relapse back into their addictive disorder or results in heightened anxiety.”

2 I agree that the way to find the article. However, as the terminology was varied for the cognitive bias. I will suggest extending the screening to find more articles. Actually, as I know, there are two articles had evaluated the memory bias and bias in response in response inhibition. (Kaohsiung J Med Sci. 2014 Jan;30(1):43-51. doi: 10.1016/j.kjms.2013.08.005. Epub 2013 Sep 14; Asia Pac Psychiatry. 2015 Jun;7(2):143-52. doi: 10.1111/appy.12134. Epub 2014 May 27.) It is not necessary to include these two articles. But, it might indicate some articles will miss in the current search way. 

We thank you Reviewer 2 for highlighting these articles. However, upon further review of these articles, they are not of relevance to our topic of interest – as we are more focused on cognitive bias. We have stated our search strategy for each of the databases. We believed these articles might have been excluded as they fulfil our exclusion criteria - no mention of any assessment or modification of cognitive biases.

3 It is good to have a table. However, the information in the table is too much. I will suggest summarising the information in the table.  It will make it more readable. 

Thank you for your recommendation. We have summarized the information we have presented within the table.

4 It is good to have a suggestion for further study. However, the suggestion should be specified. For example, it is really problems that most studies confirm the diagnosis by questionnaire. Thus, it should be discussing what problems with the diagnosis. Then, it should suggest which way to make a diagnosis is adequate. For example, how to choice diagnosis by DSM-5 or ICD-11. 

We thank you for highlighting this point. It is challenging for us to recommend whether to base the diagnosis on DSM-5 or ICD-11. We have stated that it would be preferable for studies to consider using a clinician rated questionnaire; or having a clinician to ascertain the diagnosis through a conventional psychiatric interview.

We have made the following amends:

“Most of the current research have relied on questionnaire-based assessment of whether an individual have had an underlying diagnosis of Internet addiction or gaming disorder. It might be ideal for future research to consider using either a clinician-rated questionnaire, or having a clinician to make a diagnosis based on a psychiatric interview.”

Reviewer 3 Report

The authors present an overview of studies targeting cognitive bias and cognitive bias modification in Internet Addiction and Internet Gaming Disorders. This is novel and relevant, since therapeutic interventions have mostly focused on cognitive behavioral therapy, which aims to modify cognitive biases on a conscious leve. In addition, interventons focusing on modifying automated (unconscious) cognitive processes might have the potential to have an incremental effect. 

The manuscript is well written and comprehensive. However, I have several suggestions:

1) Table 1 is very large and not reader friendly. I suggest to extract some of the contents and describe them in the text.

2) which is related to 1): the authors assume that the readers are familiar with the experimental tasks to assess attentional bias (e.g., stroop, dot probe, visual probe), with the response inhibiton tasks (e.g. go/no go) and the cognitive bias modification tasks (approach avoidance task). This might not always be the case. I would suggest to rescribe all the experimental tasks and their goals in the text, e.g., in section 3.3. This could also reduce the text in Table 1.

3) I would suggest to highlight even more the association of evidence showing cognitive bias and the potential of cognitive bias modification for therapeutic interventions

4) I would suggest to add teh keywords "Inetrnet Addiction" and "Internet Gaming Disorder"

Author Response

We thank you for your positive comments and for recognizing the importance of our work.

The manuscript is well written and comprehensive. However, I have several suggestions:

Table 1 is very large and not reader friendly. I suggest to extract some of the contents and describe them in the text.

We acknowledge this and we have summarized the contents within the table to make it more reader friendly.

which is related to 1): the authors assume that the readers are familiar with the experimental tasks to assess attentional bias (e.g., stroop, dot probe, visual probe), with the response inhibiton tasks (e.g. go/no go) and the cognitive bias modification tasks (approach avoidance task). This might not always be the case. I would suggest to rescribe all the experimental tasks and their goals in the text, e.g., in section 3.3. This could also reduce the text in Table 1.

This is a good suggestion and we have made the changes in Section 3.3.

3) I would suggest to highlight even more the association of evidence showing cognitive bias and the potential of cognitive bias modification for therapeutic interventions

We have already highlighted the association of the evidence showing cognitive bias in two of the paragraphs in the discussion. They are as follows:

“. Of the six identified studies, five have provided evidence documenting the presence of cognitive biases in these disorders. One study has reported that bias modification was effective in reducing bias.”

“Most of the studies (five out of the six) identified have reported the presence of cognitive biases in their sampled participants, except for a single study.”

As our review is a scoping review and we are limited to a qualitative synthesis of the articles, we are unable to include any additional statistics to demonstrate the strength of the association.

4) I would suggest to add teh keywords "Inetrnet Addiction" and "Internet Gaming Disorder"

Thank you for your suggestion. I have appended the additional keywords.

Round 2

Reviewer 2 Report

It is accepted to be pusblished.